# Multi-Objective Optimization of Smallholder Apple Production: Lessons from the Bohai Bay Region

**Shan Jiang, Hongyan Zhang \*** **, Wenfeng Cong, Zhengyuan Liang, Qiran Ren, Chong Wang, Fusuo Zhang and Xiaoqiang Jiao**

Department of Plant Nutrition, National Academy of Agriculture Green Development, China Agricultural University, Beijing 100193, China; sy20193030374@cau.edu.cn (S.J.); wenfeng.cong@cau.edu.cn (W.C.); zyliang1994@gmail.com (Z.L.); renqiran1220@gmail.com (Q.R.); wangchong@cau.edu.cn (C.W.); zhangfs@cau.edu.cn (F.Z.); xqjiao526@cau.edu.cn (X.J.)
\* Correspondence: zhanghy@cau.edu.cn

**Abstract:** Transforming apple production to one with high yield and economic benefit but low environmental impact by improving P-use efficiency is an essential objective in China. However, the potential for multi-objective improvement for smallholders and the corresponding implications for horticultural practices are not fully appreciated. Survey data collected from 99 apple producers in Quzhou County of Bohai Bay Region were analyzed by the Pareto-based multi-objective optimization method to determine the potential of multi-objective improvement in apple production. With current practices, apple yield was 45 t ha$^{-1}$, and the economic benefit was nearly 83,000 CNY ha$^{-1}$ but with as much as 344 kg P ha$^{-1}$ input mainly from chemical fertilizer and manure. P gray water footprint was up to 27,200 m$^3$ ha$^{-1}$ due to low P-use efficiency. However, Pareto-optimized production, yield, and economic benefit could be improved by 38% and 111%, respectively. With a concurrent improvement in P-use efficiency, P gray water footprint was reduced by 29%. Multi-objective optimization was achieved with integrated horticultural practices. The study indicated that multi-objective optimization could be achieved at a smallholder scale with realistic changes in integrated horticultural practices. These findings serve to improve the understanding of multi-objective optimization for smallholders, identify possible constraints, and contribute to the development of strategies for sustainable apple production.

**Keywords:** apple production; multi-objective optimization; phosphorus; smallholders

## 1. Introduction

In China, apples are one of the important cash crops, having a vital role in income generation for growers [1]. For fruit growers, income from their orchards can represent half of their total family income [2]. China is the largest apple producer in the world, producing 39.2 Mt in 2018, representing 45% of global production [3]. However, apple production in China faces great challenges with the industry predominately based on smallholder production [4]. One of the great challenges for apple production in China is low nutrient-use efficiency [5]. Due to knowledge, labor, and infrastructure limitations, smallholders adopt an insurance rather than a precision strategy for nutrient management, especially P management, in order to achieve high apple yield and economic benefit. It was estimated that chemical P input in the apple production of China was three times more than in developed countries in the 2010s [6–8].

With this approach, a large amount of P has accumulated in orchard soils. It will be a time-bomb for waterbodies because most orchards are located in sloping fields and have a high risk of water eutrophication due to soil erosion [9]. Besides environmental impacts and P-use efficiency,

for smallholders, apple production was closely associated with economic benefits and yield [1]. Excessive P supply often induced a reduction of apple quality, which deteriorated the economic benefits of apple production [10,11]. Therefore, facilitating apple growers to pursue multi-objectives (economic benefits, yield, P-use efficiency, and environment risks), rather than one or two objectives, is key for sustainable apple production, that is, with high economic benefit, high yield, and lowered environmental impact through improved P-use efficiency.

A series of measures have been developed to improve the sustainability of apple production. For example, orchard floor management, pollination management, and tree training systems have been employed to improve apple yield [12,13]. Furthermore, fertigation has been used to improve P-use efficiency by matching soil P supply and apple P demand both temporally and spatially [6,14]. Based on the leaf color, a nutritional status assessment for apple trees has been developed to optimize chemical P input [1]. Following the plant-soil interaction principle, local P supply in the active root zone was used to stimulate P uptake by apple trees [15]. These studies provided a valuable approach to sustainable apple production. However, most studies in apple production have been conducted with only one or two key objectives and often with single horticulture practice [16]. For smallholder growers, a focus on multi-objectives with integrated horticulture practices rather than one or two objectives with a single horticulture practice is more likely to lead to sustainable apple production.

Given the array of land preparation, weed, and pest control, nutrient management imperatives achieving sustainable apple production is complicated and involves significant changes to horticultural practices and resource allocation [17]. These changes and their desired outcomes are subject to both high expectations and significant constraints. Compared to cereal production, apple production is considerably more demanding with higher financial risks and management complexities [1]. Surveys of smallholder growers revealed that the vast majority (90%) were primarily concerned about economic benefit and yield [18]. However, for the broader society, increasingly the concern is sustainable apple production, with high produce quality and low environmental damage [19]. The lack of multi-objective achievement is a consequence of the lack of horticultural practices employed by smallholder growers from the systematic perspective and the participation of other stakeholders [20]. There are of great important compromises and potential synergies between multi-objectives for smallholder growers [21]. However, the potential of multi-objective achievement by smallholder growers and the corresponding changes in horticultural practices from the systematic perspective in apple production has not been adequately demonstrated. This potential and corresponding systematic horticulture practices are of great importance for enabling smallholder growers to improve the sustainability of apple production.

Quzhou County, Hebei Province, is within the Bohai Sea fruit-production zone, one of the major areas for apple production in China. It is a representative county, with apple production predominately on smallholdings, with high P input, high yield, high environmental impacts, and low P-use efficiency [16]. Improved understanding of the potential of multi-objective achievement and the required horticultural practices is needed to move towards sustainable apple production. Therefore, the objectives of this study were (1) to characterize the economic benefit, yield, P-use efficiency, and environmental impact of smallholder apple production in Quzhou County, and (2) to explore the potential for multi-objective achievement using Pareto-based optimization methods, and the required changes to integrated horticultural practices for smallholder apple production.

## 2. Materials and Methods

### 2.1. Survey Location and Data Collection

An intensive survey of apple growers was conducted in Quzhou County, Hebei Province (Figure 1). Quzhou County is a representative, predominantly agricultural county located on the North China Plain. It is located in one of the top three apple-producing regions of China. The average annual temperature is 13.1 °C, ranging from −10 to 30 °C. the average annual rainfall is 556 mm, ranging from 400 to 600 mm. Quzhou County has calcareous soils (pH is 7.5) with about 15.2 g kg$^{-1}$ of organic

matter, Olsen-P of 15 mg kg$^{-1}$, and total N of 1.1 g kg$^{-1}$. The soil and climatic conditions are highly suitable for apple production.

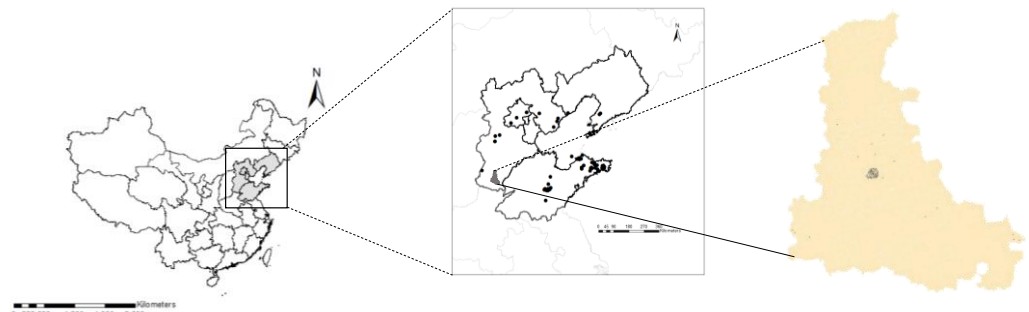

**Figure 1.** Location of data collected in this study. The gray area is the Bohai Bay region. 108 case studies about apple production were collected from published and unpublished literature in the Bohai Bay region. In addition, ninety-nine smallholder growers were surveyed in Quzhou County, a typical apple production County in the Bohai Bay region. Among these, 84 cases were collected from Xianggongzhuang Village.

In 2018, a survey of apple production practices was conducted on a 1-km grid in the whole Quhzou County. At the same time, an intensive survey was conducted in Xianggongzhuang Village, a typical apple-production village in Quhzou County. In this village, apple production is one of the major income sources for families; 53.3% of the available arable land in the village was used for apple production. Ninety-nine questionnaires were conducted by interview in the whole county, eighty-four from Xianggongzhuang Village. The structure of the 99 farms, covering the size, management forms, capital, and labor intensity, is described in Table S1. The data collected covered horticultural practices and the performance of apple production. Horticultural practices included cultivars, tree age, chemical fertilizer use, cost, amount of irrigation, and the use of reflective films, pesticides, and growth regulators. Performance data include yield and economic benefits. The method of calculating part of the production factors of chemical P fertilizer (PFP-P) and P gray water footprint was from the previous studies [8,22]. PFP-P was a useful index for sustainable apple production because it does not require measurement of yield without P fertilizer use and P uptake of the apple. It was also part of overall integrated crop management and site-specific nutrient management [23].

*2.2. P Flow in Apple Orchards*

P flow in apple orchards was calculated with P input including P from chemical fertilizer ($P_{CF}$), manure ($P_{man}$), irrigation ($P_{water}$), and deposition ($P_{dep}$):

$$P_{input} = P_{CF} + P_{man} + P_{water} + P_{dep} \tag{1}$$

where, $P_{CF}$ was calculated as the amount of chemical fertilizer use multiplied by the concentration of P in the fertilizer. $P_{man}$ was calculated as the amount of manure used multiplied by its P concentration. The amount of chemical fertilizer P and manure use was obtained from the survey data. The information of chemical fertilizer P was obtained from the fertilizer bag labels. The concentration of manure P was estimated from Chen et al., 2018. P from irrigation water and deposition was obtained from previous studies [24,25]. Here, 1.1 and 0.5 kg P ha$^{-1}$ were from irrigation water and deposition, respectively.

P output included P in the harvested apples ($P_{apple}$), P uptake by leaf and tree growth ($P_{tree + leaf}$), and P lost by leaching ($P_{leach}$), runoff ($P_{runoff}$), and soil accumulation ($P_{acc}$):

$$P_{output} = P_{apple} + P_{tree + leaf} + P_{acc} + P_{leach} + P_{runoff} \tag{2}$$

where, $P_{apple}$ was calculated as apple yield multiplied by P concentration in apples. $P_{leaf}$ and $P_{tree}$ were calculated with leaf and tree growth multiplied by their P concentration. The apple yield, and biomass of leaf and tree were obtained from the survey data. The typical P concentration in apples, tree, and leaf was obtained from previous studies [26].

$P_{leach}$ was calculated as P input ($P_{input}$) multiplied by the coefficient of P leaching in the region.

$$P_{leach} = P_{input} \times \beta_{apple} \tag{3}$$

where, $\beta_{apple}$ was 0.278% [8].

$P_{runoff}$ was calculated as P input ($P_{input}$) multiplied by the coefficient of P runoff in the region.

$$P_{runoff} = P_{input} \times \alpha_{apple} \tag{4}$$

where, $\alpha_{apple}$ was 0.03% [8].

$P_{acc}$ was calculated as the difference between P input and P output.

$$P_{acc} = P_{input} - P_{output} \tag{5}$$

### 2.3. P Gray Water Footprint

P gray water footprint ($P_{footprint}$) was calculated as:

$$P_{footprint} = (P_{runoff} + P_{leaching})/(C_{TPmax} - C_{TPnation}) \tag{6}$$

where $C_{TPmax}$ is the maximum concentration of P in a waterbody, i.e., 0.2 mg $L^{-1}$. $C_{TPnation}$ is the reference concentration in a waterbody, i.e., 0 mg $L^{-1}$ [8].

### 2.4. Economic Benefit and Partial Factor Production of Chemical P Fertilizer

Economic benefit was calculated as the difference between the cost of all material input, (including fertilizers, irrigation, and labor) and the output of apple:

$$E_{benefit} = C_{apple} - C_{fertilizer} - C_{manure} - C_{irrigation} - C_{labor} - C_{pesticide} - C_{herbicide} - C_{film} \tag{7}$$

All values were obtained from the survey data.

Partial factor production of chemical P fertilizer (PFP-P) was calculated as the ratio of apple yield and chemical P fertilizer use, expressed as kg apple per kg $P_2O_5$.

### 2.5. Pareto-Based Multi-Objective Optimization

The trade-offs between yield, economic benefits, P-use efficiency, and P gray water footprint were explored with multi-objective Pareto-based ranking. The exploration of the trade-offs between objectives was formulated with the 99 samples as follows:

$$Max\ U(x) = (U_1(x), U_2(x) \ldots U_k(x))^T$$

$$X = (x_1, x_2 \ldots x_n)^T \tag{8}$$

Subject to i constraints:

$$g_i(x) \le h_i \tag{9}$$

where, $U_1(x) \ldots U_k(x)$ are the objective functions that are simultaneously maximized or minimized, and $(x_1 \ldots X_n)$ are the decision variables that represent adjustable parameters to identify the potential horticultural practices. Constraints in Equation (9) can arise from the problem formulation, for instance by limitations on farm model results related to a specific configuration of decision variables. In the

present study, apple yield, PFP-P, and economic benefit should be simultaneously higher than current practices (CPs), which were above 45 t ha$^{-1}$, 72 kg kg$^{-1}$ P$_2$O$_5$, and 82,846 CNY ha$^{-1}$, respectively. P gray water footprint should be below 27,212 m$^3$ ha$^{-1}$.

The first criterion for the performance of a case study is its Pareto rank as proposed by Goldberg (1989). Individuals in the population are Pareto-optimal when they do not perform worse than any other individual for all of the objectives. That is, these perform equal to or better than any other individual in at least one objective. In such cases, there is no objective basis to discard the individual. These individuals are called non-dominated and receive a rank of 1. In the present study, Pareto rank 1 cases were selected as optimal growers (OPT). Except for the OPT growers, the rest of the growers were selected as current practices (CPs). This set of solutions is called the trade-off frontier. The next step in Pareto-ranking the entire population of solutions is to remove the individuals of rank 1 from the population and identify a new set of non-dominated individuals, which is assigned rank 2. This process is continued until all individuals in the population are assigned a Pareto rank. When information on the prior performance of the farming system is used, the ranking mechanism of Goldberg (1989) may be slightly adjusted to improve the selection of that part of the solution space where solutions are found that perform better than the original practices. In this case, a (superior) rank 0 is assigned to solutions that perform better than the original configuration for all the objectives.

### 2.6. Data Analysis

A *t*-test was performed with the SAS statistical software (SAS Inst., Cary, NC, USA). Significant difference among means was determined by LSD at $p \leq 0.05$. Data was presented by Sigma-Plot (Version 12.0, Systat Software Inc., San Jose, CA, USA) and e!Sankey (version 4.1, Hamburg, Germany).

## 3. Results

### 3.1. P Inputs and P Flow in Orchard

With current practice (CP), total P input was as high as 344 kg ha$^{-1}$, 87% from chemical P fertilizer with apple yield of 45 t ha$^{-1}$, containing 9.3 kg P, being only 2.7% of the total P input (Figure 2a). As much as 86% of the applied P remained in the soil with 5.7 kg P ha$^{-1}$ lost by leaching and runoff. Compared with CP, total P input at the Pareto-based optimal (OPT) was reduced by 40%, most of the reduced P was from chemical P fertilizer. Accumulated P in soil was reduced to 156 kg ha$^{-1}$. Soil P leaching and runoff were reduced by 28% (Figure 2b).

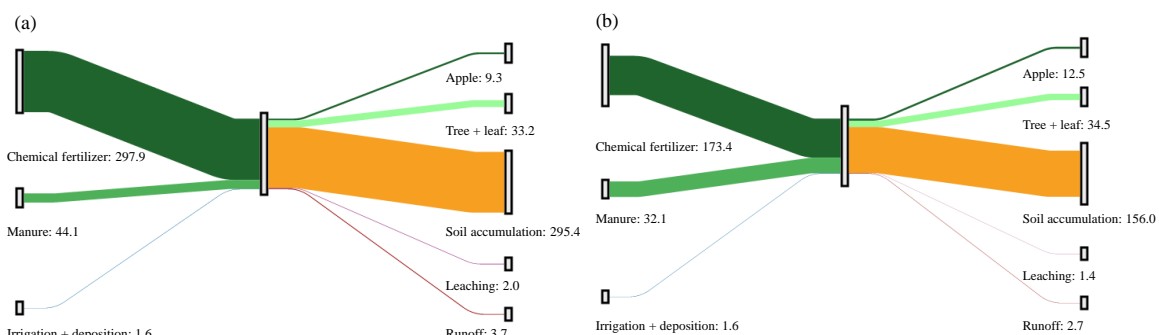

**Figure 2.** P flux (kg P ha$^{-1}$) of apple orchard under current grower practice (CP) (**a**) and for Pareto rank 1 growers (OPT) (**b**) in Quzhou County in 2017. Ninety-nine smallholder growers were surveyed in Quzhou County and Xianggongzhuang Village. Five OPT growers were selected with Pareto-based optimization approach.

### 3.2. Performance Indicators in Grower Practices and Pareto-Based Optimal Growers

The exploration of trade-offs between the four objectives with Pareto-based optimization showed that changes in horticultural practices could lead to substantial improvements in yield and economic

benefit and PFP-P and reduced P gray water footprint (Figure 3). Synergies were apparent between PFP-P with yield, as well as economic benefit, respectively (Figure 3b,d). A similar trend was observed between apple yield and economic benefit (Figure 3f). High PFP-P was associated with higher economic benefit and yield but with low negative environmental impact. Trade-offs were found between P gray water footprint and PFP-P (Figure 3a) since horticultural practices with low P-use efficiency often have high environmental P impacts. There was no significant relationship between P gray water footprint and economic benefit, as well as apple yield (Figure 3c,e).

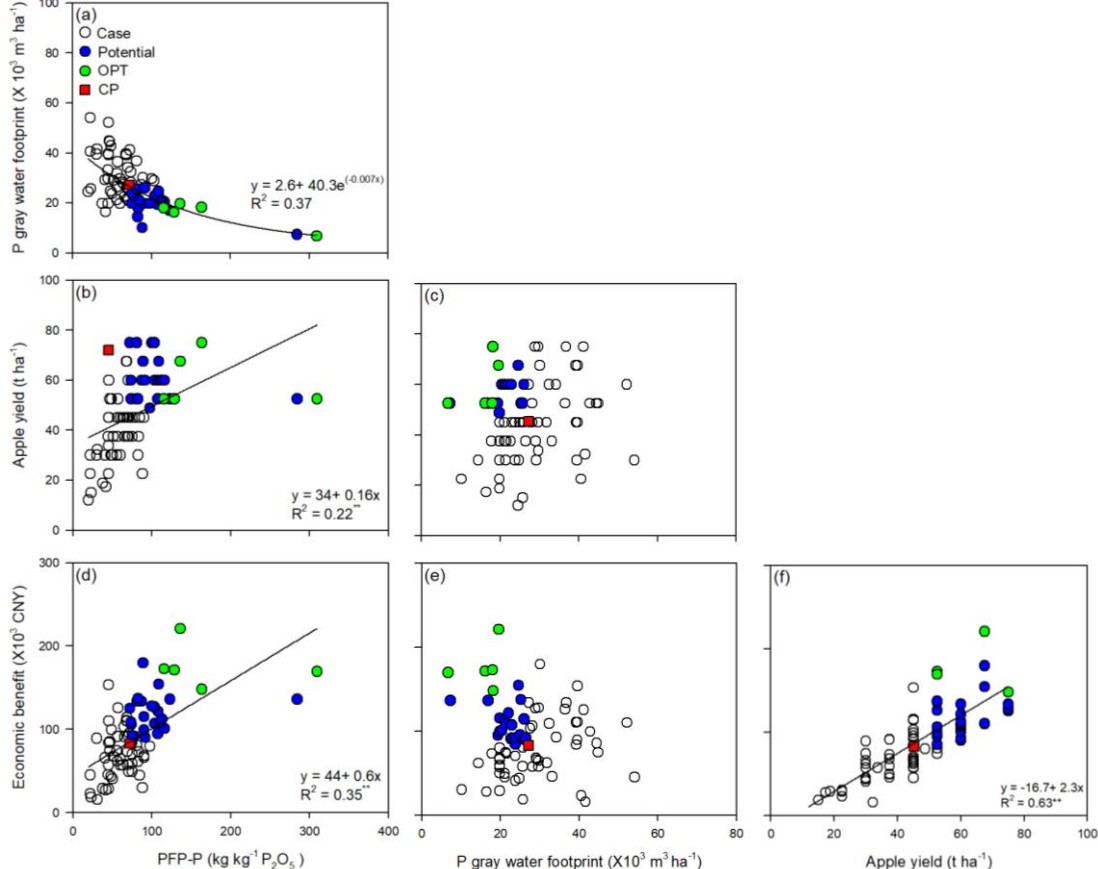

**Figure 3.** Relationships between performance indicators in apple orchards of smallholder growers in Quzhou County in 2017.These were represented by Pareto frontiers after multi-objective optimization for the full exploration of the frontier trade-off frontier. Each dot represents a performance configuration, green symbols for Pareto rank 1 growers (OPT), and blue for solutions that outperform the original solution in all objectives (Potential). The red symbols represent current farmer practice (CP). The white symbols represent each surveyed case study (Case). Here, PFP-P was defined as kilograms of grain produced per kilogram $P_2O_5$ applied.

As indicated in Figure 3, five growers (green symbols) performed better than others in trade-offs between the four objectives and were set as OPT. Other growers (blue symbols) were also better than current practice, suggesting a considerable potential to improve the trade-offs between the four objectives.

### 3.3. Four Objectives and Corresponding Integrated Horticulture Practices

Compared with CP, apple yield in OPT was 38% higher, P-use efficiency was improved by 116%, economic benefit was improved by 111%, and P gray water footprint was reduced by 29% (Figure 4). The integrated measures for high apple yield and P-use efficiency were employed by growers in OPT

(Figures 5 and 6). Chemical P fertilizer use was reduced by 42%, and the total cost of all inputs was reduced by 32%, whereas the proportion of reflective film increased by 23%. A similar farm size in CP and OPT was obtained.

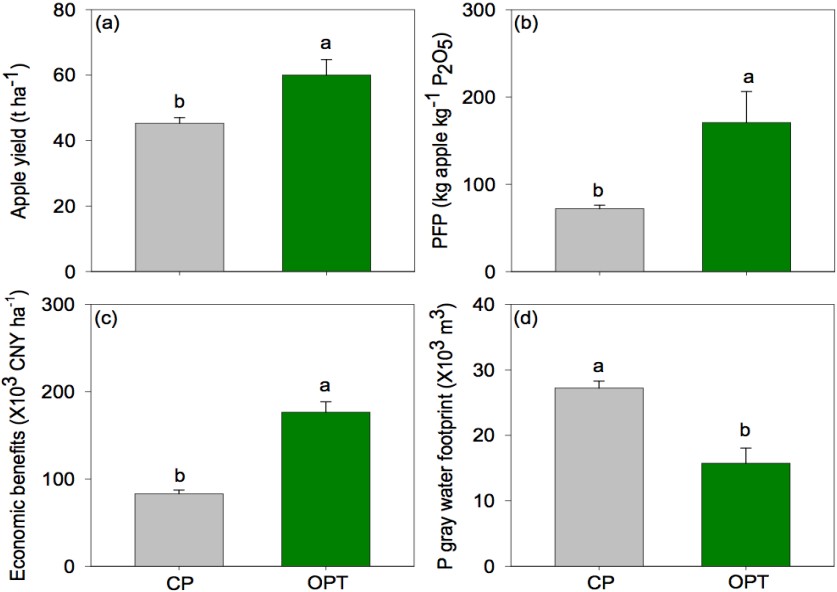

**Figure 4.** Apple yield (**a**), partial factor of productivity (PFP), defined here as kilograms of apples produced per kilogram of $P_2O_5$ applied (**b**), economic benefit (**c**), and P gray water footprint (**d**) under current practice and OPT growers in Quzhou County in 2017. Each value is the mean of cases (+SE). Different lower case letters denote significant difference ($p \leq 0.05$) between categories.

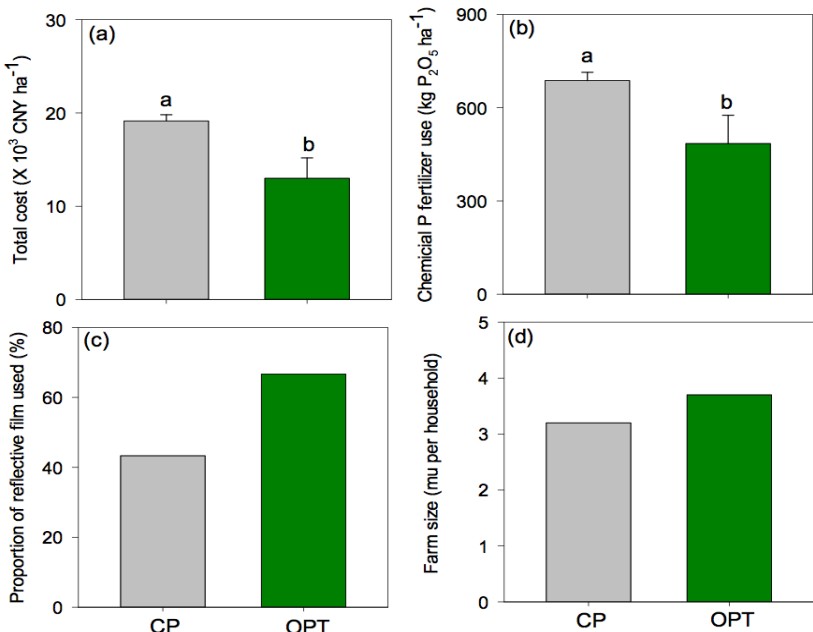

**Figure 5.** Total cost (**a**), P application rate (**b**), proportion of reflective film used by growers (**c**), and farm size (**d**) under current farmer practice (CP) and Pareto rank 1 growers (OPT) in Quzhou County in 2017. In (**a**) and (**b**), each value is the mean of cases (+SE). Different lower case letters denote significant difference ($p \leq 0.05$) between categories.

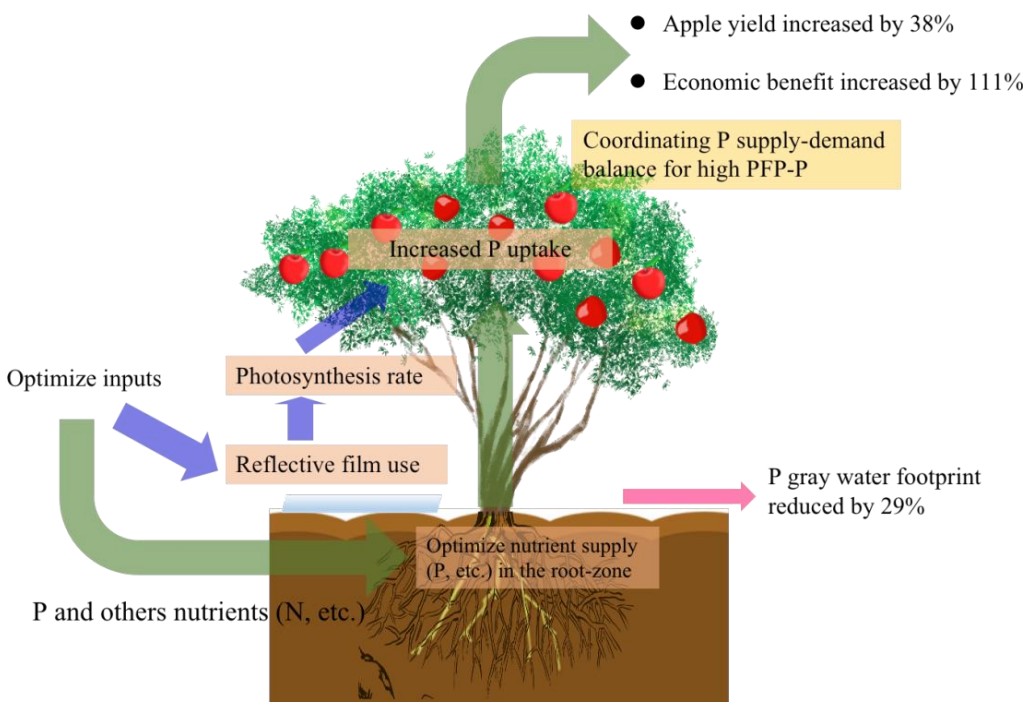

**Figure 6.** Concept model of multi-objective achievement in apple production by smallholder growers in the North China Plain. Corresponding integrated horticulture practices were presented.

### 3.4. Characteristics of Five Solutions Near the Extremes (Minima or Maxima) of the Four Objectives

The complexity of the orchard with multiple objectives was represented by Pareto-based optimization. With CP, apple yield was 45 t ha$^{-1}$, and PFP-P was only 72 kg kg$^{-1}$ P$_2$O$_5$. Most of chemical P applied accumulated in topsoil, contributing to high P gray water footprint (27,212 m$^3$ ha$^{-1}$). Economic benefit was only 82,846 CNY ha$^{-1}$. Based on the horticultural practices employed by growers, the best performing growers for each of the objectives are presented (Table 1). The contribution of each horticultural practice to the objective value was estimated by changing the value of the horticultural practices and expressing the resulting change in the objective value relative to the original value.

When aiming for maximum yield a considerable improvement from 45 to 75 t ha$^{-1}$ would be possible. The adaption of increasing reflective film and herbicide use could be an effective approach to make it. These changes also lead to 116.3% improvement in P-use efficiency, 111% increase in economic benefit and 29% reduction in P gray water footprint, compared with CP. Maximizing economic benefit could be achieved by considerable reductions in chemical P fertilizer, reflective film, pesticides, and growth regulators. Maximizing P-use efficiency and P gray water footprint was associated with a marked reduction in chemical P fertilizer use.

**Table 1.** Characteristics of six solutions near the extremes (minima or maxima) of the four objectives (apple yield, PFP, economic benefit, and P gray water footprint) and its corresponding horticulture practices.

| Items | Variable | CP | Highest Apple Yield | | Highest PFP | | Highest Economic Benefit | | Lowest P Gray Water Footprint | |
|---|---|---|---|---|---|---|---|---|---|---|
| | | | Value | Δ (%) | Value | Δ (%) | Value | Δ (%) | Value | Δ (%) |
| Objectives | Apple yield (t ha$^{-1}$) | 45.3 | 75 | | 52.5 | | 67.5 | | 52.5 | |
| | PFP (kg apple kg$^{-1}$ P$_2$O$_5$) | 72 | 122.4 | | 309.6 | | 136.3 | | 309.6 | |
| | Economic benefit (×10$^3$ CNY ha$^{-1}$) | 83 | 158 | | 169 | | 221 | | 169 | |
| | P gray water footprint (m$^3$ ha$^{-1}$) | 27,212.9 | 17342.6 | | 6718.8 | | 19,617.8 | | 6718.8 | |
| Horticultural practices | Chemical N use (kg N ha$^{-1}$) | 822.6 | 803 | −2.4 | 366.5 | −55.4 | 505.4 | −38.6 | 366.5 | −55.4 |
| | Chemical P use (kg P ha$^{-1}$) | 211.5 | 301.5 | 42.6 | 74.1 | −65.0 | 216.3 | 2.3 | 74.1 | −65.0 |
| | Chemical P fertilizer cost (CNY ha$^{-1}$) | 3745.2 | 3763.1 | 0.5 | 924.7 | −75.3 | 2699.9 | −27.9 | 924.7 | −75.3 |
| | Total cost of chemical fertilizer (CNY ha$^{-1}$) | 19,149.2 | 14,917.6 | −22.1 | 8810.3 | −54.0 | 20,831.15 | 8.8 | 8810.3 | −54.0 |
| | Cultivated area (mu) | 3.2 | 6.8 | 112.5 | 3.2 | 0.0 | 3 | −6.3 | 3.2 | 0.0 |
| | Tree age (years) | 13.8 | 12.5 | −9.4 | 27 | 95.7 | 11 | −20.3 | 27 | 95.7 |
| | Proportion of growth regulators used (%) | 6.8 | 0 | −6.8 | 0 | −6.8 | 0 | −6.8 | 0 | −6.8 |
| | Proportion of reflective transfer film used (%) | 43.4 | 100 | 56.6 | 100 | 56.6 | 0 | 43.4 | 100 | 56.6 |
| | Proportion of herbicide used (%) | 40.5 | 50 | 9.5 | 0 | 40.5 | 0 | 40.5 | 0 | 40.5 |

Values displayed relate to objectives and decision variables for horticultural practices based on a plant-soil interaction principle. Partial factor of productivity (PFP), defined here as kilograms of apple produced per kilogram P$_2$O$_5$ applied. Δ (%) indicated the relative change of horticulture practices for each extreme of objectives, compared with current practice (CP). The underline values were the magnitude of the improvement for the performance of objectives.

## 4. Discussion

### 4.1. Trade-Off between Objectives

One distinctive feature of the results is synergies between PFP-P and apple yield, as well as economic benefit (Figure 3b,d). Concurrently, a trade-off between PFP-P and P gray water footprint was observed (Figure 3a). This suggests that PFP-P is the key factor for sustainable apple production and underpins a potential strategy to produce high economic benefit and yield with limited negative environmental impacts by improving P-use efficiency. PFP-P is complex with many components. Here it is defined as yield produced per unit of $P_2O_5$ applied. In practice, PFP-P has always been part of overall integrated crop management and site-specific nutrient management due to the relationship between yield and P input [27].

In contrast, previous studies showed that high yield is accompanied by high P input, and it often results in low PFP [28,29]. Many factors contributed to the low PFP, one of the key factors is excessive chemical P use in apple production [1,7]. However, it is widely recognized that applying large amounts of chemical P fertilizers is not a cost-effective way to increase apple yield because yield responses to chemical P fertilizer use were negligible when soil Olsen-P concentration in the orchard was high [30,31]. However, fertilizers in China are readily available to most growers [32]. In addition, due to a lack of nutrient diagnostic tools for apple production, in order to pursue high-yielding and economic benefit, growers often adopt an insurance strategy, rather than a precision approach, to apply chemical P based on their empirical knowledge [33,34]. With this approach, large amounts of chemical P should not be used by crops in the first growing season as this results in low economic benefit and high negative environmental impact due to low PFP [35]. In addition, chemical P accumulation in croplands can cause a deficiency of Ca, Mg, and S and reduce apple quality and economic benefit due to the antagonism between P and other elements in the soil [36].

However, in the present study, a positive relationship between PFP-P and apple yield, as well as economic benefit, was observed (Figure 3b,d). This shows that high-yielding, economic benefit, and high PFP-P in apple production can be simultaneously achieved. The key value of this finding is a way to improve P-use efficiency in apple production. Many attempts have been made to improve P-use efficiency (PFP-P) for sustainable crop production [37,38]. For instance, in P-deficiency soil, yield response approach was widely used. The method describes the crop-fertilizer response in different soil fertility conditions and helped growers to secure crop yield and to optimize P use at a large scale, but environmental impact has been largely neglected [30]. In high soil P conduction, a building-up and maintenance approach was developed considering P balance and soil P conditions. For instance, with this approach, chemical fertilizer P could be reduced by 20% and P-use efficiency improved by 30% in cereal crops [30]. Recently, maximizing the root biological potential of P acquisition, as a strategy to improve P-use efficiency, has been advocated by many researchers [38,39]. Compared with broadcast application, P-use efficiency increased 1.5 times due to root growth induced by localized P application in apple production [39]. Furthermore, by applying fertigation through a drip irrigation system, the P-use efficiency of apple trees can be substantially improved [40].

For a certain apple yield level, to improve PFP-P is often associated with low P loss to the environment. In the present study, a trade-off between PFP-P and P gray water footprint was obtained (Figure 3a). The key point of improving PFP-P in crop production is to match the soil P supply in the root zone to crop demand spatially and temporally [37,41]. This needs a better understanding of the required P uptake by apple trees and the required chemical P fertilizer application for a certain yield increase to minimize P loss to the environment. One of the major factors affecting PFP is apple yield. Therefore, the recommended P application in high-yielding and high P-use-efficiency apple production should consider yield. For instance, based on apple demand for P, 120–150 kg $P_2O_5$ ha$^{-1}$ has been recommended to achieve an annual yield of 30 t ha$^{-1}$, with 10% P recovery efficiency in the first growing season with limited P loss [30]. In addition, P-use efficiency has been improved by 10% due to the application of fertilizer formulas developed considering soil P supply and apple P

demand [7]. However, in general, these studies focus on one or two objectives, but multiple objectives in apple production for smallholders will be urgently needed.

*4.2. Achieving Multi-Objective Apple Production by Smallholder Growers*

The core aim of this paper is to demonstrate the potential for achieving multi-objective of apple production by smallholder growers and its corresponding integrated horticulture practices with a Pareto-based ranking approach. Thereby providing a strong indication that smallholder growers can manage trade-offs to deliver high yield and economic benefit, high P-use efficiency, and lowered environmental risk in a multiple-win model by changing integrated horticulture practices. In the present study, 99 smallholder growers in apple production were surveyed, and five growers with optimal performance were identified by the Pareto-based optimization approach (Figure 3). Compared with current practice, OPT growers could produce 38% more apples with 111% higher economic benefit and 29% reduction in P gray water footprint by a 116% improvement in PFP (Figure 4). The key point of corresponding integrated horticultural practices, including chemical P use, high-yielding practices, and reduced total costs, would need to be employed by smallholder growers (Figures 5 and 6). With this approach, sustainable apple production (high yield and economic benefit with limited P-related environmental footprint by improving P-use efficiency) with integrated sound horticultural practices is a realistic alternative for smallholder growers.

Specifically, this approach provided a clear overview and understanding of integrated horticultural practices employed by smallholder growers for sustainable apple production [42]. It could be used as a tool within a visioning and back-casting exercise. With this approach, the desirable or best-performing growers will be identified, it will encourage stakeholders to participate in scientific research, and it allows stakeholders to be more creative in considering how to overcome barriers limiting the desirable outcomes, rather than to discuss the real problems and barriers to change. In the present study, in order to achieve high PFP apple production without compromising other objectives, an integrated horticulture practice was proposed. For instance, chemical P fertilizer would be reduced from 298 kg ha$^{-1}$ in CP to 212 kg ha$^{-1}$ in OPT (Figure 2). Similar research has been conducted to optimize landscape use with multi-objective under sets of control variables, and diverse agronomy practices have been provided [43]. An earlier study showed that high-yielding maize with high nutrient-use efficiency and limited negative environmental impact could be achieved by improved agronomic practices [44]. This provided the sound technology needed to achieve sustainable crop production. However, most studies have focused on a single technology or had limited objectives. Furthermore, intensive field trails and in situ measurements will be needed to validate predictions. For smallholder growers, an integrated, systematic, and holistic approach, rather than a single technology, is needed to achieve multi-objective apple production.

Therefore, in the present study, combined crop and nutrient management has been employed by Pareto-based optimization growers. Compared with CP, apple yield under OPT was 63 t ha$^{-1}$. This is higher than the average yield in the USA (27.8 t ha$^{-1}$), France (44.0 t ha$^{-1}$), Germany (25.4 t ha$^{-1}$), and Italy (40.1 t ha$^{-1}$) [3]. However, there is still a large yield gap compared to yield potential. With individual objective optimization, apple yield in Pareto-optimal was as high as 75 t ha$^{-1}$. In order to achieve the highest apple yield without compromising other objectives, some horticulture practices will be employed. However, there are many yield-limiting factors for apple production, such as tree age, mulch [45,46], crop load [47], and rootstocks [48]. In the present study, combined reflective film and other inputs was an effective approach to improving yield (Table 1). Previous studies showed that reflective film can increase leaf nutrient content and increase leaf photosynthetic capacity and thereby, significantly improve fruit yield and quality [16,49]. With this approach, the P output of apple production increased substantially. Reflective transfer films and herbicides should be used in the right manner because they would cause serious environmental impact, such as degraded soil, water pollution by runoff, and leaching [50,51]. However, few works of literature have reported P-related environment footprint induced by reflective transfer films and herbicide use in apple production.

In addition to crop management, P management strategy was also employed by OPT growers. Compared with CP, chemical P fertilizer use in OPT was reduced to 212 kg P ha$^{-1}$ (Figure 2). This is still higher than in developed countries (45–90 kg P ha$^{-1}$) [52]. When maxima PFP-P was achieved, chemical P fertilizer use was 74 kg P ha$^{-1}$, which is in the range of developed countries (Table 1). This indicated that there is a clear potential to reduce chemical P fertilizer use to improve P-use efficiency. In order to produce enough food to feed a large population, large amounts of chemical P have been applied into cropland, which reduced P use efficiency substantially and induced a series of environmental risks [28]. With this approach, as high as 56 Mt of P was accumulated in China's major cropland, and it is a time-bomb for water eutrophication [29]. About 1.5 Mt of P will be lost to waterbody each year by runoff due to excessive chemical P use in China [25]. It was estimated that half of the chemical P applied could be saved without any yield loss in China, which is equivalent to the amount of chemical P used in Europe [29].

In high-yielding apple production, high PFP-P is often associated with lower environmental impacts. In the present study, a negative relationship was obtained between PFP and P gray water footprint (Table 1, Figure 4). Optimization of chemical P fertilizer use is an effective approach to improve PFP and reduce environmental impact. Chemical fertilizer use reduced from 687 kg P$_2$O$_5$ ha$^{-1}$ to less than 485 kg P$_2$O$_5$ ha$^{-1}$. This requires an intensive knowledge-based management strategy for sustainable apple production to be adopted by smallholder growers. For example, the application of liquid phosphate fertilizer through fertigation can significantly improve crop P nutrition and reduce the fixation and adsorption of P in soil and P losses to the environment [53,54]. Apple production was strongly linked with environment and economic domains. With adaptive horticulture practices, achieving high yield and PFP and limited environmental impact is often associated with high economic benefits. In the present study, the total cost, including chemical fertilizers, pesticides, and irrigation, was reduced with optimal practice (Table 1). Among all of the inputs, the cost of chemical fertilizer was the greatest cost (50%). The economic benefit could be improved by optimized nutrient management [34].

Apple production involves many practices, including land preparation, chemical fertilizer use, pesticide use, and herbicide use. For smallholder growers, it is not easy to achieve the optimal approach with multi-objective optimization due to the limitations of labor, knowledge, and infrastructure and the complexities of apple production. To achieve sustainable apple production, smallholder growers will be required to operate in their orchards as precisely as scientists do in field experiments [55]. It is a knowledge-intensive process. Even the OPT growers in the present study accumulated as much as 156 kg P ha$^{-1}$ in orchard soils, which is equivalent to the annual P consumption in a wheat-maize rotation [56]. This indicates that more research is needed to improve P use in apple production. The study demonstrated the potential of multi-objective achievement for apple production and adaptive integrated horticulture practices were identified. However, making this a reality in the orchard and transferring knowledge to smallholders needs considerable further joint research engaged with scientists and other stakeholders (growers, etc.). At the same time, a commitment to effective extension will urgently need to upscale the adoptive integrated horticulture practices into a large scale. This will require a systematic, holistic, and interdisciplinary approach, rather than a one-size-for-all approach, to resolve the problems that will be encountered by smallholder growers [57,58]. Furthermore, some effective and contextual policies should be made to encourage the engagement of scientists and other stakeholders (smallholder growers, etc.) to develop adaptive technologies based on the knowledge of multi-objective optimization. Only with robust and urgent cooperation between scientists, enterprises, and policymakers, we can find effective solutions to achieve sustainable apple production from the grass-root level.

## 5. Conclusions

Compared with current practice, yield and economic benefit under multi-objective optimized apple production could be improved by 38% and 111%, respectively. P gray water footprint could

be reduced by 29% due to improved P-use efficiency. Integrated horticulture practices (combined optimal chemical P input, adaptation of reflective films, and other cost-effective approaches) can be employed to achieve multi-objective apple production. In addition, the multi-objective optimization approach demonstrated the potential for sustainable apple production. These results indicated that multi-objective optimization could be achieved by smallholder apple growers by modifying integrated horticultural practices. The integrated horticultural practices and implications addressed in this study have potential applicability in other regions facing similar challenges.

**Supplementary Materials:** The following are available online at http://www.mdpi.com/2071-1050/12/16/6496/s1, Table S1: Description and summary statistics for all variables at baseline used in the study.

**Author Contributions:** Conceptualization, F.Z. and H.Z.; methodology, X.J.; software, Z.L.; validation, W.C.; formal analysis, S.J. and Q.R.; investigation, S.J. and X.J.; resources, W.C. and C.W.; data curation, S.J. and Q.R.; writing—original draft preparation, S.J.; writing—review and editing, X.J.; supervision, H.Z. and C.W.; project administration, F.Z. All authors have read and agreed to the published version of the manuscript.

**Funding:** This work was supported by National Key R&D Program of China (2017YFD0200200/0200206), National Natural Science Foundation of China (NSFC) (31701999), Science and Technology Talents and Platform Program of Yunnan Province (2019IC026), Yunnan Yuntianhua Co., Ltd. (YTHZWYJY2018001) and China Scholarship Council (No.201913043).

**Conflicts of Interest:** The authors declare no conflict of interest.

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
