# Peer review of "Multi-Objective Optimization of Smallholder Apple Production: Lessons from the Bohai Bay Region"

_sustainability, doi:10.3390/su12166496_

Round 1

Reviewer 1 Report

The paper is well-written, interesting, and touches important problems of horticultural production in China. Although I feel that when discussing ‘sustainable apple production' more environmental aspects should be taken into account/approached, the paper surely addresses some important challenges and gives some insight into the possibilities and potential for increasing the performance and sustainability of apple production in China.

I suggest some minor corrections, as specified below:

Abstract vs. Introduction: In the abstract (line 19) it is written that P input = 344 kg ha-1, while in the introduction (line 39) it is written that P input can be as high as 238 kg ha-1.

Line 39: 238 kg P kg ha-1 – should be 238 kg P ha-1

Lines 42-43: “…and easy to erosion to cause water eutrophication” – this sentence needs to be clarified/corrected.

Line 47 rather THAN one or two objectives…

Line 48: I would suggest to avoid using the word “industry” when you write about the sustainable horticultural production.

Line 75: are OF great importance.

Line 81: I would suggest to remove “industry”.

Line 98: WERE collected (instead of was). And LITERATURE (instead of LITERATURES).

Lines 102-104: English language/grammar correction of the sentence is necessary.

Line 106: eighty-four in…

Line 107-108: The data collected covered horticultural practices and performance of apple production and growers – not clear what is meant here (performance of growers).

Line 108: ages -> age

Line 109: the use OF reflective films.

Line 110: ‘partial factor production…’ - language correction of the sentence is necessary.

Line 112-113: ‘it does not need to measure yield without P fertilizer 112 use and P uptake of apple’ - language correction of the sentence is necessary.

Line 121: INFORMATION was obtained (and not the concentration of fertilizer). It should be revised.

Line 144: ‘maximize’ -> maximum.

Line 193-194: ‘mostly them reduced’ – sentence needs to be corrected.

Line 231: P2O5 – 5 should be in subscript (and not superscript).

Figure 6: ‘P and others’ should be rather ‘P and other nutrients’.

Table 1: Title of the table should be revised. “Characteristics of six solutions near the extremes…” does not explain what is in the table (table should be self-explanatory). Most of the text from the title should be presented in the footnote of the table. Table could be re-formatted to fit to one page (which could be done easily, without compromising the content). ‘tree ages’ -> tree age.

Lines 264-265: ‘(…) The adaption of increasing reflective film and herbicide use could be an effective approach to make it. (…)’ – what about the ‘environmental sustainability’ aspect of using reflective film and increase the herbicides use? This should be discussed in the paper.

Line 298: ‘was widely use’ -> was widely used.

Line 300: ‘environment impact’ -> environmental impact

Line 303: ‘feeding roots not soil…’ – sentence needs to be corrected/clarified.

Line 333-334: ‘(…) sustainable apple production (high yield and economic benefit with limited environment impacts (…)’ – it should be made clear that only P-related aspect of environmental footprint of the apple production has been taken into account in this study. So the authors should not conclude on the overall ‘limited environmental impact’.

Line 336: ‘understanding integrated…’ -> understanding of integrated.

Lines 361-363: past tense is and present tense is used (increase, improved) – it should be unified/used appropriately.

Line 372-373: I would suggest to replace the word “penalty” with other expression. ‘(…)which is equivalently…’ – should also be corrected (language).

Line 376: ‘optimize’ -> optimization of.

Line 386: ‘optimize’ -> optimized.

Line 406: ‘integrated’ – start a new sentence with a capital letter.

Lines 414-419: The text of this section should not be put in the quotation marks; authors’ initials rather than full names should be used.

Line 422: full stop at the end of the sentence.

Author Response

Responses to the reviewers’ comments

We are appreciative of the constructive criticisms and comments offered by the reviewers and yourself on the manuscript. An extensive revision as suggested by the reviewers has been finished. The detailed point-by-point responses to the reviewers’ comments are listed in the following section.

Reviewer #1:

The paper is well-written, interesting, and touches important problems of horticultural production in China. Although I feel that when discussing ‘sustainable apple production' more environmental aspects should be taken into account/approached, the paper surely addresses some important challenges and gives some insight into the possibilities and potential for increasing the performance and sustainability of apple production in China.

Response: Thank you very much for your constructive suggestions. more environmental aspects should be taken into account in sustainable apple production. It was related with GHG emission, water eutrophication and other issues. In the present study, the research zone was focused on Bohai Bay Region. It is one of the three apple production zone in China. Most orchards are located in sloping field. What is serious is that smallholder growers often apply more chemical P fertilizer than apple demand, leaving huge amount of P accumulated in orchard. Therefore, in the present study, P-related environment footprint was highlighted. We added more words about environmental impact induced by chemical P use. Please see the line 384-386.

I suggest some minor corrections, as specified below:

Abstract vs. Introduction: In the abstract (line 19) it is written that P input = 344 kg ha-1, while in the introduction (line 39) it is written that P input can be as high as 238 kg ha-1.

Response: On average, as high as 238 kg P ha-1 was used by apple growers in China [1]. But there are great variations, ranging from 152 kg P ha-1 to 491 kg P ha-1. In the present study, the total P input was 344 kg P ha-1, and 298 kg P ha-1 was from chemical P fertilizer. It was in the range of previous studies [2]. We have changed the sentence as “It was estimated that chemical P input in apple production of China was three times more than in developed countries in the 2010s”. Please see the line 39-41.

References:

[1] Zhao, Y.; Li, Z.; Liu, M.; Xiao, X.; Wang, C.Y.; Sun, D.F.; Lun, F. Phosphorus budgets and their associated environmental risks in the main apple orchard areas in China from 2006 to 2016. J. Agro-Environ. Sci. 2019, 38, 2779-2787. (in Chinese with English abstract)

[2] Zhu, Z.; Xia, Y.; Liu, J.; Ge, S.; Jiang, Y. Analysis of Soil Phosphorus Input and Phosphorus Environment Load Risk in Major Apple Production Regions of Shandong Province. Acta Hortic Sinica. 2017, 44, 97-105.

Line 39: 238 kg P kg ha-1 – should be 238 kg P ha-1

Response: We have changed the sentence as “It was estimated that chemical P input in apple production of China was three times more than in developed countries in the 2010s”. Please see the line 39-41.

Lines 42-43: “…and easy to erosion to cause water eutrophication” – this sentence needs to be clarified/corrected.

Response: We have changed the sentence as “most orchards are located in sloping fields and have high risk of water eutrophication due to soil erosion”. Please see the details in line43-44. 

Line 47 rather THAN one or two objectives.

Response: We revised it as “rather than one or two objectives”. Please see the details in line 49.  

Line 48: I would suggest to avoid using the word “industry” when you write about the sustainable horticultural production.

Response: Yes, we agree. We have revised it as “…is key for sustainable apple production, that is, with high economic benefit, high yield and lowered environmental impact through improved P use efficiency.” Please see the details in line 49-51.  

Line 75: are OF great importance.

Response: We revised it as “There are of great important compromises…”. Please see the details in line 73.  

Line 81: I would suggest to remove “industry”.

Response: Yes, we agree. We have changed the sentence as “Improved understanding of the potential of multi-objective achievement and the required horticultural practices is needed to move towards sustainable apple production.” Please see the details in line 83.  

Line 98: WERE collected (instead of was). And LITERATURE (instead of LITERATURES).

Response: We revised it as “…were collected…”. And “literatures” has been changed as “literature”. Please see the details in line 99.  

Lines 102-104: English language/grammar correction of the sentence is necessary.

Response: We revised these sentences. “At the same time, an intensive survey was conducted in Xianggongzhuang Village, a typical apple production village in Quhzou County. In this village, apple production is a major source of income for families.  53.3% of the available arable land in the village was used for apple production”. Please see the details in line 104-107.

Line 106: eighty-four in..

Response: We revised it as “…by interview in Whole county, eighty-four in Xianggongzhuang village.” Please see the details in line 110.

Line 107-108: The data collected covered horticultural practices and performance of apple production and growers – not clear what is meant here (performance of growers).

Response: We have deleted the “performance of growers”. The manuscript highlighted the sustainable and systematic technologies of multi-objective apple production. The characteristics of growers was not considered in the manuscript, although it is very important for apple growers to adopt these technologies. We will conduct intensive research about this topic in the further. Please see the details in line 113. 

Line 108: ages -> age

Response: We revised it as “Horticultural practices included cultivated areas, tree age, …” Please see the details in line 114.

Line 109: the use OF reflective films.

Response: We revised it as “…chemical fertilizer use, cost, amount of irrigation, and the use of reflective films, …”. Please see the details in line 114.

Line 110: ‘partial factor production…’ - language correction of the sentence is necessary.

Response: We revised it as “The method of calculating part of the production factors of chemical P fertilizer (PFP-P) and P gray water footprint were from the previous studies [8, 22].” Please see the details in line 115-117.

Line 112-113: ‘it does not need to measure yield without P fertilizer 112 use and P uptake of apple’ - language correction of the sentence is necessary.

Response: We revised it as “PFP-P was a useful index for sustainable apple production, because it does not require measurement of yield without P fertilizer use and P uptake of apple.” Please see the details in line 119.

Line 121: INFORMATION was obtained (and not the concentration of fertilizer). It should be revised.

Response: We revised it as “Information of chemical fertilizer P was obtained …”. Please see the details in line 129.

Line 144: ‘maximize’ -> maximum.

Response: We changed it from maximize to maximum. Please see the details in line 151.

Line 193-194: ‘mostly them reduced’ – sentence needs to be corrected.

Response: We revised it as “most of the reduced P was from chemical P fertilizer.”. Please see the details in line 200-201.

Line 231: P2O5 – 5 should be in subscript (and not superscript).

Response: We revised it in line 239. 

Figure 6: ‘P and others’ should be rather ‘P and other nutrients’.

Response: We revised it in Figure 6.

Table 1: Title of the table should be revised. “Characteristics of six solutions near the extremes…” does not explain what is in the table (table should be self-explanatory). Most of the text from the title should be presented in the footnote of the table. Table could be re-formatted to fit to one page (which could be done easily, without compromising the content). ‘tree ages’ -> tree age.

Response: We revised the table title as “Characteristics of six solutions near the extremes (minima or maxima) of the four objectives (apple yield, PFP, economic benefit, and P gray water footprint) and its corresponding horticulture practices in apple production.” Presented the left text into the footnote. Please see the details in line 264-265.  

We have changed “tree ages” as “tree age” in the table

The table has been re-formatted to fit to one page.

Lines 264-265: ‘(…) The adaption of increasing reflective film and herbicide use could be an effective approach to make it. (…)’ – what about the ‘environmental sustainability’ aspect of using reflective film and increase the herbicides use? This should be discussed in the paper.

Response: Previous studies showed that reflective film can increase leaf nutrient content, and increase the leaf photosynthetic capacity and thereby, significantly improve fruit yield and quality [1]. With this approach, the P output of apple production increased substantially. However, reflective transfer films and herbicides should be used in a right manner, because they would cause serious environment impact, such as degraded soil, water pollution by runoff and leaching [2,3]. P-related environment footprint induced by reflective transfer films and herbicides use in apple production was not fully understood. Please see the details in line 372-375.

References:

[1] Zhang, Z.; Wang, L.; Gao, J.; Chen, Q.; Wang, S.; Zhang, C. A preliminary report on the effect of moisture permeability reflective film on crown illumination and fruit quality of sweet cherry trees. China Fruits. 2019, 3, 52-56. (in Chinese)

[2] Bellec, F.L.; Velu, A.; Fournier, P.; Squin, T.L.; Michels, T.; Tendero, A.; Bockstaller, C. Helping farmers to reduce herbicide environmental impacts. ECOL INDIC. 2015, 54, 207-216.

[3] Wang, L.; Tian, L. Study on the prevention and treatment of tapetum lucidum pollution in Orchard of Yantai city. Yantai Fruits2019, 01, 5-6. (in Chinese)

Line 298: ‘was widely use’ -> was widely used.

Response: We have changed it as “was widely used”. Please see the details in line 305.

Line 300: ‘environment impact’ -> environmental impact

Response: We have changed it as “environmental impact”. Please see the details in line 307.

Line 303: ‘feeding roots not soil…’ – sentence needs to be corrected/clarified.

Response: We revised it as “Recently, to maximize root biological potential of P acquisition, as a strategy to improve P use efficiency, has been advocated by many researchers”. Please see the details in line 310-311.

Line 333-334: ‘(…) sustainable apple production (high yield and economic benefit with limited environment impacts (…)’ – it should be made clear that only P-related aspect of environmental footprint of the apple production has been taken into account in this study. So the authors should not conclude on the overall ‘limited environmental impact’.

Response: We revised it as “With this approach, sustainable apple production (high yield and economic benefit with limited P-related environmental footprint by improving P use efficiency) with integrated sound horticultural practices is a realistic alternative for smallholder growers.”. Please see the details in line 341-342.

Line 336: ‘understanding integrated…’ -> understanding of integrated.

Response: We have changed as “…a clear overview and understanding of integrated horticultural practices employed by smallholder growers…”. Please see the details in line 344-345.

Lines 361-363: past tense is and present tense is used (increase, improved) – it should be unified/used appropriately.

Response: We revised it as “Previous studies showed that reflective film can increase leaf nutrient content, and increase the leaf photosynthetic capacity and thereby, significantly improve fruit yield and quality [16,54]”. Please see the details in line 369-371.

Line 372-373: I would suggest to replace the word “penalty” with other expression. ‘(…)which is equivalently…’ – should also be corrected (language).

Response: We revised it as “It was estimated that half of the chemical P applied could be saved without any yield losses in China, which is equivalently with the amount of chemical P use in Europe [29]”. Please see the details in line 385-386.

Line 376: ‘optimize’ -> optimization of.

Response: We have changed it as “Optimization of chemical P fertilizer use is an effective approach…”. Please see the details in line 390-391.

Line 386: ‘optimize’ -> optimized.

Response: We have changed it as “optimized”. Please see the details in line 400.

Line 406: ‘integrated’ – start a new sentence with a capital letter.

Response: We have changed it as “Integrated”. Please see the details in line 426.

Lines 414-419: The text of this section should not be put in the quotation marks; authors’ initials rather than full names should be used.

Response: We deleted the quotation marks and abbreviated names. Please see the details in line 434-438.

Line 422: full stop at the end of the sentence.

Response: We added the full stop. Please see the details in line 446-447.

Reviewer 2 Report

The research question is adequately expressed and satisfied. I recommend only a few specific arguments to be discussed by authors. First, the true nature of the problem  is the trade-off between environmental and economic sustainability. Three of the 4 objectives (i.e. economic benefit, apple yield and PFP-P) can be merged in one (Maybe economic benefits or PFP-P), making ridundant the other two. Your results trivially confirm this expected condition. Please, specify why did you choose to complicate the analysis. Second, the Liebig principle, and the resulting diminishing returns law, is a well accepted principle that can be read in any agronomy handbook for dummies. Please, start from this principle before defining the terms of the problem. Third, nothing is said about the structure of the 99 sample farms: are their sizes, management forms, capital and labor intensity highly differentiated? This is important when describing the farms belonging to different ranks of the Pareto scale. Consequently, fourth, please analyze the true motives preventing the use of the maximum FPP-P practices by smallholders. And suggest normative solutions (eg, taxes on fertilizers, credit support for precision fertirrigation).

Author Response

Responses to the reviewers’ comments

We are appreciative of the constructive criticisms and comments offered by the reviewers and yourself on the manuscript. An extensive revision as suggested by the reviewers has been finished. The detailed point-by-point responses to the reviewers’ comments are listed in the following section.

Reviewer #2:

The research question is adequately expressed and satisfied. I recommend only a few specific arguments to be discussed by authors.

Response: Thanks a lot for your suggestions.

First, the true nature of the problem is the trade-off between environmental and economic sustainability. Three of the 4 objectives (i.e. economic benefit, apple yield and PFP-P) can be merged in one (Maybe economic benefits or PFP-P), making redundant the other two. Your results trivially confirm this expected condition. Please, specify why did you choose to complicate the analysis.

Response: Thanks a lot for your constructive suggestions. Generally speaking, the true nature of sustainable apple production is the trade-off between environmental and economic sustainability. In the specific case in the Bohai Bay Region, we should consider the demand of different stakeholders in apple production in detail under the two pillars (environmental and economic sustainability) of sustainable apple production. Surveys of smallholder growers revealed that the vast majority (90%) were primarily concerned about economic benefit. For the broader society, high PFP-P in apple production should be put on the top priority due to the limited phosphate reserve. For China government, high apple yield is highlighted in order to provide sufficient food to meet the large populations’ demand. In addition, in order to pursuit of high yield and economic benefits in apple production, apple growers often apply more chemical P fertilizers than apple demand, leaving large amount of P accumulated in orchard. It will be a time-bomb for waterbodies, because most orchards are located in sloping fields and have high water eutrophication risk due to soil erosion. From the environmental perspective, limited environmental impact induced by chemical P fertilizer use in apple production should be highlighted. Therefore, in the study, four objectives were selected to evaluate multi-objective optimization of apple production in the Bohai Bay Region.

Second, the Liebig principle, and the resulting diminishing returns law, is a well accepted principle that can be read in any agronomy handbook for dummies. Please, start from this principle before defining the terms of the problem.

Response: Thanks a lot for your constructive suggestions. Frankly speaking, I have tried my best to catch the starting point of this suggestion. Unfortunately, I failed. I wonder whether you can kindly say more about it in detail. In the study, trade-offs between apple yield, economic benefits, P use efficiency (PFP-P), and P gray water footprint were explored with multi-objective Pareto-based ranking. At the same time, the required changes to integrated horticultural practices for sustainable apple production was explored based on the principle of plant-soil interactions. All the indicators (apple yield, economic benefits, P use efficiency (PFP-P), and P gray water footprint), obey the Liebig principle, and the resulting diminishing returns law. We have provided some potential solutions to achieve multi-objective apple production by smallholder growers. In order to achieve one objective of apple production, it is easy to identify the critical level under some constraints. However, the optimal level for multi-objective achievements of apple production was not fully understood. In the future, more research will be needed to address this issue.

Third, nothing is said about the structure of the 99 sample farms: are their sizes, management forms, capital and labor intensity highly differentiated? This is important when describing the farms belonging to different ranks of the Pareto scale.

Response: Thanks a lot for your constructive suggestions. Yes, the structure of the 99 sample farms should be made clear in the manuscript. We have added a table in the supporting information about the structure of the 99 sample farms. These farms are managed by smallholders. Most of apple growers are old and not well educated. In general, most apple growers work in the orchard 5 hours per day during the apple growing periods. However, they got very little information about sustainable apple production. On average, the total family income per household was 100,000 CNY per year, most of them are from apple production. Due to lack of available knowledge and technology for sustainable apple production by growers, there are great variations about the size, management forms, capital and labor intensity. On average, the farm size we surveyed is 0.22 ha, ranging from 0.03 ha to 0.67 ha. Horticultural practices, including cultivars, tree age, chemical fertilizer use, cost, amount of irrigation, and the use of reflective films, pesticides and growth regulators, were employed by apple growers. Inappropriate horticulture practices employed by growers were quite common. For instance, most growers believe that high input always comes with high yield. Excessive chemical P fertilizer use was very common. On average, 298 kg P ha-1 was used by growers, while only 43 kg P ha-1 was used by apple, resulting low P use efficiency (PFP-P) and great P accumulation in orchard. Please see the details in line 111-112 and supporting information (Table S1).

Table S1 Description and summary statistics for all variables at baseline used in the study.

Variables

Description

Mean

SD

Growers' age (year)

Ages of orchard manager

54.3

9.2

Education level (year)

Primary education is 6; Secondary education is 9; High school education is 12

8.1

3.3

Cultivated area (ha)

The area of apples planted

0.2

0.1

Tree age (years)

Growing years since planted

13.7

6.5

Chemical N use (kg N ha-1)

The amount of chemical N use per unit area

920.1

485.6

Chemical P use (kg P ha-1)

The amount of chemical P use per unit area

293.4

103.4

Proportion of reflective transfer film used (%)

1=YES; 0=NO

0.4

0.5

Proportion of herbicide used (%)

1=YES; 0=NO

0.4

0.5

Apple yield (t ha-1)

Total apple production per unit of area

46.6

15.4

Total cost (×103 CNY)

All cost in apple production

22.7

5.4

Economic benefit (×103 CNY)

Profit of apple production

89.8

44.6

Observation

99.0

Consequently, fourth, please analyze the true motives preventing the use of the maximum PFP-P practices by smallholders. And suggest normative solutions (eg, taxes on fertilizers, credit support for precision fertirrigation).

Response: Thanks a lot for your constructive suggestions. PFP-P was a useful index for sustainable apple production, because it does not require measurement of yield without P fertilizer use and P uptake of apple. However, it was also a complex index for smallholders, because it links the part of overall integrated crop management and chemical P management. Maximum PFP-P practices are closely related with apple yield and amount of chemical P fertilizer input. In order to achieve high PFP-P by apple growers, they have to maximize apple yield and minimize chemical P fertilizer input. It requires apple growers to think about it from systematic and holistic perspective. Knowledge about high PFP-P apple production from the systematic and holistic perspective was not fully understood by scientists based on the plant-soil interactions. Most research were conducted to develop “one size fits all” technologies [1]. The effective products and high PFP-P technology, especially based on the requirement of multi-objective apple production was not fully developed due to lack of effective dialogue between scientists and apple growers. Scientists often assume apple growers will automatically apply technologies developed by them because of the formal logic from single indiscipline [2]. Apple growers have very limited interests in participating in scientists’ research due to lack of interdisciplinary research. The poor scientist-farmer engagement often leads to PFP-P practices that prescribe action instead of facilitating learning.

In the present study, PFP-P in highest PFP-P solution was 4.3-fold of that in CP (Table 1). This requires adaptive technologies of high PFP-P in apple production and engagement of scientists and apple growers. First, the scientists need to understand the basic principle of soil P supply and apple P uptake temporally and spatially from interdisciplinary perspective. Secondly, adaptive technologies of high PFP-P should be developed by engagement of scientists and apple growers in the research process. It should be involved multidisciplinary knowledge. The last and most important, in order to be accepted and applied by apple growers in large scale, some policies, for instance, taxes on fertilizers, made by governments should consider multi-objective achievement with multidisciplinary knowledge [3]. Please see the details in line 412-422.  

References:

[1] Kanter, D.; Musumba, M.; Wood, S.; Palm, C.;Antle, J.; Balvanera, P.; Dale, V.; Havlik, P.; Kline, K.; Scholes, R.; Thornton, P.; Tittonell, P.; Andelman, S. Evaluating agricultural trade-offs in the age of sustainable development. Agric. Syst. 2016, 73-88.

[2] McCown, R.; Changing systems for supporting farmers" decisions: problems, paradigms, and prospects. Agric. Syst. 2002, 74, 0-220.

[3] Bryan, B.A.; Kandulu, J.M. Designing a policy mix and sequence for mitigating agricultural non-point source pollution using deliberative multi-criteria evaluation. Water Resour. Manag. 2011, 25, 875-892.
